# Spray Flame Synthesis and Multiscale Characterization of Carbon Black–Silica Hetero-Aggregates

**DOI:** 10.3390/nano13121893

**Published:** 2023-06-20

**Authors:** Simon Buchheiser, Ferdinand Kistner, Frank Rhein, Hermann Nirschl

**Affiliations:** Process Machines, Institute of Mechanical Process Engineering and Mechanics, Karlsruhe Institute of Technology, 76131 Karlsruhe, Germany; ukxmk@student.kit.edu (F.K.); frank.rhein@kit.edu (F.R.)

**Keywords:** hetero-aggregation, small-angle X-ray scattering, carbon black, silica, spray flame, nanoparticle characterization

## Abstract

The increasing demand for lithium-ion batteries requires constant improvements in the areas of production and recycling to reduce their environmental impact. In this context, this work presents a method for structuring carbon black aggregates by adding colloidal silica via a spray flame with the goal of opening up more choices for polymeric binders. The main focus of this research lies in the multiscale characterization of the aggregate properties via small-angle X-ray scattering, analytical disc centrifugation and electron microscopy. The results show successful formation of sinter-bridges between silica and carbon black leading to an increase in hydrodynamic aggregate diameter from 201 nm to up to 357 nm, with no significant changes in primary particle properties. However, segregation and coalescence of silica particles was identified for higher mass ratios of silica to carbon black, resulting in a reduction in the homogeneity of the hetero-aggregates. This effect was particularly evident for silica particles with larger diameters of 60 nm. Consequently, optimal conditions for hetero-aggregation were identified at mass ratios below 1 and particle sizes around 10 nm, at which homogenous distributions of silica within the carbon black structure were achieved. The results emphasise the general applicability of hetero-aggregation via spray flames with possible applications as battery materials.

## 1. Introduction

The urgent need for a reduction in the use of fossil fuels in order to reduce ${\mathrm{CO}}_{2}$ emissions has led to the emergence of electro mobility, e.g., in the form of electric cars. Currently, lithium-ion batteries are the most frequently used component for energy storage. The batteries consist of multiple components, including active materials, polymer binders and carbon black. Therein, the active materials are responsible for energy storage, and the carbon black increases the conductivity via the creation of electrical pathways between the active material particles. At the cathode side, a frequently used binder is polyvinylidene fluoride (PVDF), which ensures the uniformity and binding of the active materials, correct slurry rheology [1] and the dispersion of the carbon black [2]. However, PVDF is part of the group of per- and polyfluoroalkyl substances (PFASs), which are responsible for the persistent and bioaccumulating pollution of the environment during production [3] and recycling [4]. Thus, the European Union is evaluating a possible ban on the production, use, sale and import of PFASs until September 2023, which is why the development and evaluation of alternatives is of the utmost importance. However, PVDF has the advantages of a high mechanical strength, corrosion resistance and chemical stability, making it difficult to replace [5]. One option is to improve the properties of the carbon black network with another additive to reduce the technical requirements on the binder. In polymers, the addition of silica to carbon black is used to adjust mechanical properties. One main application is the reinforcement of tires [6], in which the miscibility of both components is linked to durability [7]. Although silica has insulating properties, it is able to improve the generation of electrical pathways via carbon black, which may lead to increased conductivities [8]. Recently investigated carbon–silica structures for possible applications in Li-ion batteries as anode materials include hollow spheres [9], aerogels [10], amorphous powders [11] and carbon-coated silica nanoparticles [12]. To overcome the limitations of volume expansion due to lithiation at the anode, the silica is nanostructured [13,14]. Moreover, on the cathode side, nanostructured silica has been shown to improve the electrochemical cycling stability [15]. Regarding the desired structure of carbon black, an open, high-surface-area structure [16] that forms a robust carbon black–binder network is important for the electrochemical performance [17,18]. Therein, the aggregate size of the carbon black has a significant impact on the carbon black–binder network, with smaller particles (<250 nm) forming a more cross-linked network [19]. The current methods for the synthesis of carbon black–SiO_2_ composite materials include complex manufacturing methods that are not easily scalable for industrial production. Therefore, this work introduces a hetero-aggregation process carried out via the sintering of silica nanoparticles directly onto the carbon black aggregates in a spray flame. Spray flames are scalable [20] and allow for the dispersion of both particles due to the active mixing zone above the nozzle [21]. The feasibility of the spray flame synthesis of hetero-aggregates has been shown frequently for catalysts in which the sintered contact between the catalytic material and promoter is able to improve the catalytic activity [22,23]. For hetero-aggregates of carbon black and silica, the sintered hetero-contact may lead to improvements in mechanical strength and dispersibility. In theory, this leads to a reduction in the demand on the multi-functionality of the polymeric binder, opening up alternatives to PVDF. Another possible improvement gained via hetero-aggregation is an increase in aggregate stability, leading to easier processing during dry mixing, during which conductivity due to the deformation and breakage of carbon black is possible [24]. 

In order to achieve the desired macroscopic properties, the hetero-aggregation process and its influencing factors must be understood on a microscopic level. However, the resulting hetero-aggregates are highly fractal nanoscale structures that consist of two different amorphous materials and are therefore difficult to reliably characterize. Hence, a multiscale characterization ranging from primary particle properties to aggregate properties is needed. Common characterization methods for hetero-aggregates are laser light diffraction [25,26], X-Ray diffraction [26] and electron microscopy [27,28]. In order to obtain comprehensive information about primary particles, as well as aggregate and mixing properties, a combination of ultra-small-angle X-ray scattering (USAXS), high-angle annular dark-field scanning transmission electron microscopy (HAADF-STEM) with energy-dispersive X-ray spectroscopy (EDXS) and analytical disc centrifugation (ADC) is used. USAXS provides integral information about the aggregates and is routinely used for aggregates of carbon and silica, as well as products obtained via spray flame synthesis [29,30,31,32]. The results include fractal dimensions of the aggregates as well as the mean primary particle sizes of both materials. ADC allows for the determination of changes in the hydrodynamic aggregate size for varying flame parameters. The combination of HAADF-STEM and EDXS yields qualitative information about aggregate shapes, primary particle size distributions and the homogeneity of the dispersion. In order to understand the behaviour of hetero-aggregation in a two-material system, the aggregation mechanisms of pure carbon black and pure silica particles (one-material system) are first investigated. Afterwards, the obtained findings are compared with the experimental results of the hetero-aggregation process in order to determine the influence of the primary silica particle size on the hetero-aggregate size, primary particle properties, fractal dimensions and the homogeneity of the hetero-aggregation process. As a result, optimized experimental conditions are presented.

## 2. Materials and Methods

### 2.1. Synthesis of Silica Particles

The colloidal silica particles were synthesised via the Stoeber process [33], using ammonia (25% ammonia solution for analysis, Merck, Darmstadt, Germany) as a catalyst and tetraethyl orthosilicate (TEOS) (98% tetraethoxysilane, Alfa Aesar, Karlsruhe, Germany) as a precursor. The cosolvent was ethanol (VWR Chemicals, Darmstadt, Germany). Temperature control and mixing were achieved using a magnetic stirrer with a heating plate. The exact experimental conditions are provided in the Appendix A. Table 1 lists the names of the samples with the corresponding mean particle diameter, $d_{50,0}$ +/− one standard deviation of the number size distribution averaged over a triple measurement, as determined via dynamic light scattering (Zetasizer nano ZS, Malvern Panalytical, Malvern, UK). 

### 2.2. Production of the Hetero-Aggregates

For the production of the hetero-aggregates, a carbon black of the type TIMCAL Super C65 (Nanografi Nano technology, Çankaya/Ankara, Turkey) was suspended in ethanol with a mass concentration $x_{\mathrm{CB}}$ of 0.5 weight-%. The suspension of the colloidal silica particles was added to the stock solution in varying mass ratios to the concentration of carbon black that ranged from $5\;\frac{x_{\mathrm{CB}}\;}{x_{{\mathrm{SiO}}_{2}}\;}$ to $0.2\;\frac{x_{\mathrm{CB}}\;}{x_{{\mathrm{SiO}}_{2}}}$. The particles were then dispersed with ultrasonic waves (Branson Sonifier 450, Branson Ultrasonics, Danbury, CT, USA) to break up the existing agglomerates. For clarification, in this publication, loosely bonded particles, e.g., particles bonded by Van der Waals forces, are referred to as agglomerates, whereas aggregates are considered particle networks bonded by strong interactions, e.g., solid sinter bridges. The suspension was then fed into the spray flame using a SpraySyn burner (University of Duisburg-Essen, Duisburg, Germany). The gas flows were controlled via mass flow controllers by Bronkhorst (Ruurlo, The Netherlands ). A laminar pilot flame was ignited with 2 standard litres per minute (slm) of CH_4_ and 12 slm of O_2_. To stabilise the flame, a sheath gas flow of 120 slm of pressurized air was added. The spray flame was created in the centre of the burner, where a canula with an annular gap was located. A syringe pump (Harvard Instruments, Holliston, MA, USA) conveyed the prepared suspension through the canula. The gas flow of 10 slm of O_2_ through the annular gap atomized the suspension. The resulting spray was then continuously ignited by the pilot flame, which created the spray flame. A scheme of the burner is presented in the Appendix A with an overview of the burner parameters. Further general data regarding the burner are published in [34]. At 12 cm above the burner surface, a hole-in-tube probe collected the hetero-aggregates on a nanoporous track-etched membrane (Whatman Nuclepore Track-Etched Membranes, Cytiva, Amersham, UK) with a pore diameter of 200 nm.

### 2.3. Aggregate Characterization

The hetero-aggregates produced were characterized using small-angle X-ray scattering (Xeuss 2.0 Q-Xoom, Xenocs SA, Sassenage, France) in order to obtain information about the multiple structural characteristics ranging from the primary particle diameters to the fractal dimension of the aggregate. Beam generation was achieved with the X-ray micro focus source Genix3D Cu ULC (Ultra Low divergence), which emits Cu-Kα radiation with an energy of 8.04 keV and a wavelength of 1.5406 Å. For an extended measuring range, a Bonse-Hart-module [35] for USAXS measurements was installed. The powder samples were prepared for measurement on adhesive polyimide foil. The chosen exposure time was 30 min, with a distance of 2500 mm from the sample to the detector. A comparative image analysis of the HAADF-STEM (FEI Tecnai Osiris, FEI company, Hillsboro, OR, USA) images, carried out via ImageJ [36], yielded a number-based size distribution of the primary particles. The hydrodynamic aggregate size distribution was measured with an analytical disc centrifuge (CPS instruments, Prairieville, LA, USA). Before the measurement, the particles are suspended in deionised water (0.01 mass-%) and dispersed with ultrasonic waves (Branson Sonifier 450, Branson Ultrasonics, Danbury, CT USA) for the breakage of the agglomerates. 

#### Small-Angle X-ray Scattering

The scattering of incoming X-Rays on nanoparticles is characteristic to their nanoscale structures, such as particle size and morphology. The resulting scattering curve is a double logarithmic plot of an intensity I over the scattering vector q in ${Å}^{-1}$. The scattering vector q in Equation (1) describes the scattering angle 2θ independent of the wavelength λ of the primary beam and is provided by: (1)$$q=\frac{4\pi }{\lambda }\sin\left(\theta \right).$$

Guinier’s law (Equation (2)) yields information about particle or aggregate size in the form of the radius of gyration $R_{g}$ with pre-factor G at small angles (q$R_{g}$ < 1):(2)$$I\left(q\right)=G\;\exp\left(-\frac{q^{2}R_{g}^{2}}{3}\right)$$

The slope of the scattering curve is provided by a local power law fit with prefactor B according to:(3)$$I\left(q\right)={B\;q}^{-P}.$$

For fractal aggregates, the exponent P is equal to the fractal dimension of the mass of the aggregate $D_{\mathrm{FM}}$ [37]. For surface fractals, the exponent P is also proportional to the fractal dimension of the surface $D_{\mathrm{FS}}$.
(4)$$P=6-D_{\mathrm{FS}},$$ resulting in values for P between 3 and 4, making it possible to distinguish mass and surface fractals. If the exponent P equals 4, the particles exhibit an ideally smooth surface with a sharp density transition, fulfilling Porod’s law [38]. Further evaluation of the scattering data was performed via the unified fit model (Irena Package 2.71 [39], IgorPro, Wavemetrics, Portland, OR, USA), according to Beaucage [40]. The results of the unified fit yield a log-normal size distribution with the assumption of spherical primary particles. The geometric standard deviation $\sigma_{g}$ is calculated from the parameters of Guinier’s law and the power law fit with the polydispersity index (PDI) [41]:(5)$$\sigma_{g}=\exp\left(\sqrt{\frac{\ln\left(\mathrm{PDI}\right)}{12}}\right);\;\mathrm{PDI}=\frac{{B\;R}_{g}^{4}}{1.62\;G}.$$

The mean diameter of the distribution is then calculated as follows [32]:(6)$$d_{\mathrm{SAXS}}=2\sqrt{\frac{5}{3}}R_{g}\exp(-13\frac{\ln\left(\mathrm{PDI}\right)}{24}).$$

Due to the possible convolution of scattering information from the silica and carbon black in the hetero-aggregates, the above-mentioned methodology was only applied to pure materials.

## 3. Results

### 3.1. Pure Carbon Black in the Spray Flame

Figure 1 shows the changes in the primary particle and aggregate properties of the pure carbon black in a spray flame using ethanol as dispersion liquid. Before the spray flame, the carbon black had an open fractal structure with a fractal dimension of mass of 2.2, which was derived from the scattering curve with a unified fit, as depicted in Figure 1a. The open-branched structure is also observable in the HAADF-STEM image, Figure 1d. The radius of gyration of the primary particles was $R_{g}=46.4$ nm and exhibited a sharp density transition with P=4, suggesting a smooth particle surface. From the fit parameters B=2.06 and $G=8.2\cdot 10^{9},$ a number-weighted primary particle size distribution with a median diameter of 38 nm and a geometric standard deviation of $\sigma_{g}=0.4$ was calculated using Equation 5. This particle size is in good agreement with the primary particle size reported by Spahr et al. [42]. The ADC yielded a hydrodynamic aggregate size distribution with a mode at 159 nm. After the spray flame, the hydrodynamic aggregate size of the carbon black increased to $d_{\mathrm{mod}}=201$ nm, with a reduction in the fractal dimension of mass to 1.8, suggesting further aggregation in the spray flame. In the TEM image (Figure 1e), the fractal structure of the carbon black is still observable. The analysis of the primary particles via TEM imaging resulted in a mean particle diameter of 39.6 nm with a geometric standard deviation from $\sigma_{g}=0.28$. The retrieved fit parameters of the unified fit of the scattering data after the spray flame were $R_{g}=33.3$ nm, $G=1.1\cdot 10^{10}$ and B=6.7. With P=4, the primary particle surface properties are unchanged. The calculated size distribution shows a decrease in polydispersity in the geometric standard deviation from $\sigma_{g}=0.4$ to 0.35 and a decreased primary particle diameter of 35.5 nm. The differences in absolute values are the results of the resolution limit for TEM and potential deviations from the assumptions drawn from the SAXS data. 

One major advantage of the holistic approach is the combined evaluation of multiple particle and aggregate properties. Therein, only negligible changes in the primary particle properties after the spray flame process are revealed, whereas the hydrodynamic aggregate size increased due to further aggregation. Because of the newly formed inter-aggregate connections, the structure became more open, which led to a decrease in the fractal dimension of mass. 

### 3.2. Pure Colloidal Silica in the Spray Flame

In order to evaluate the changes in the particle properties of the pure colloidal silica in the spray flame using ethanol as dispersion liquid, a comparison of the SAXS data and the TEM analysis was conducted. Figure 2 shows exemplary results for the 60 nm colloidal silica particles before and after the spray flame. In Figure 2a, the scattering curve of the sample after the spray flame exhibits multiple structure levels, which are also visible in the accompanying HAADF-STEM image, Figure 2b. For scattering vectors in the range of 0.01 Å^−1^ and 0.1 Å^−1^, the properties of the primary particles (marked in red) were retrieved. The local Guinier fit yielded a radius of gyration $R_{g}=25.3$ nm which corresponds to a sphere equivalent diameter of 65.3 nm. A comparison with the TEM data yielded a mean diameter of 61.1 nm and shows good agreement. Furthermore, oscillations characteristic for monodisperse spheres are observed in the scattering data. With a fractal dimension of surface $D_{\mathrm{FS}}=2,$ the particles exhibited a smooth surface with a spherical shape, which is identical to the properties prior to the spray flame. The silica particles visibly aggregated and formed sinter bridges in the flame, which were both observable via TEM and SAXS (marked in blue). A local power law fit results in a fractal dimension of mass $D_{\mathrm{FM}}=1.7$. In contrast, the scattering curve of the sample before the spray flame levels off, indicating the characteristic spherical shape. In consequence, no significant aggregation or other large structures were present before the spray flame. The TEM images of the sample after the spray flame revealed that most silica particles are directly connected to one to three other particles, resulting in a mean coordination number of 2.1 ± 0.9. The contact length between two aggregated silica primary particles was roughly half of the particle diameter. However, an infrequent coalescence of the silica particles, leading to the formation of large structures in the micron size range, was detected (marked with green). These structures were also observed in the USAXS data with a radius of gyration $R_{g}=367.4$ nm. The local power law fit yields an exponent P=2.8, showing an overlap between the fractal dimension of surface of the structure (P between 3 and 4) and the fractal dimension of mass of the silica aggregate, which mainly consists of the smaller primary particles (P=1.7). 

In conclusion, the colloidal silica particles aggregated heavily in the spray flame. Although coalescence was observed, most primary particles preserved their spherical shape. Their infrequent coalescence into larger structures may be attributed to the residence time distribution of the particles in the turbulent spray flame. 

### 3.3. Characterization of the Hetero-Aggregates

The findings obtained for the spray flame experiments of the pure materials were also observed for the hetero-aggregates of carbon black and silica. In Figure 3a, a HAADF-STEM image of a hetero-aggregate with visible hetero-contact is shown. Closeups of the hetero-contact between the carbon black and silica are depicted in Figure 3b–d. Due to sintering, a large contact area between the carbon black and silica was formed. A clearly defined border between the two materials is detected within the resolution limit. 

As shown in Section 3.2, silica–silica homo-contacts and coalescence may lead to the formation of homo-aggregates consisting of pure silica. The formation of these homo-aggregates is visible in the SAXS data shown in Figure 4a. The slope at ultra-small angles ($q<10^{-3}{\;Å}^{-1}$) was fitted with a power law fit (Equation (3)), and the exponent P was evaluated for different original particle sizes of silica and for different mass ratios of silica to carbon black in Figure 4b. Similar to the data shown in Figure 2a the exponent P is influenced both by the fractal dimension of mass of the aggregate (P∼1.8−2.1 for carbon black) as well as the fractal dimension of surface of the homo-aggregate (P=4 for colloidal silica). Consequently, a higher exponent corresponds with a higher relative degree of homo-aggregation. In the scattering data, two major trends were observed: on one hand, the exponent P is influenced by the particle size of the silica. For the hetero-aggregates produced with 10 nm and 40 nm silica particles with mass ratios of one, the exponent P was close to the fractal dimension of mass of the pure carbon black aggregates, with values typically between 2.2 and 2.3 (see Section 3.1). For hetero-aggregates produced at the same mass ratio but with silica particles 60 nm in size, the exponent P is increased to 2.8. On the other hand, the mass concentration of silica in the hetero-aggregates significantly influences homo-aggregation. For a silica particle size of 40 nm, the exponent increased from 1.8 for a mass ratio of 0.33 to 2.5 for a mass ratio of 3. The HAADF-STEM image in Figure 4c shows the formation of an irregular structure due to coalescence accompanied by homo-aggregated 30 nm silica primary particles for a mass ratio of 3 and thus supports the findings acquired via the SAXS analysis. For this particle size, an increase from 1.9 (mass ratio 0.33) up to 2.5 emphasizes the importance of choosing lower mass ratios in order to avoid homo-aggregation and possible coalescence. 

The trends observed in the USAXS measurements are further supported by the ADC measurements depicted in Figure 5. For mass ratios above or equal to three, the PSD is bimodal (Figure 5a). The first mode is close to the observed hydrodynamic aggregate size of pure carbon black, while the second mode corresponds to a structure in the micron size range, possibly the homo-aggregates. The relative weight of the second mode increases with a decrease in the silica particle size. Even though the formation of homo-aggregates is observed, the carbon black aggregates were incorporated into the formed silica structures for smaller particle sizes. This was especially evident for the 10 nm silica particles at a mass ratio of 3 for which only a single mode in the micron size range is measured. Therefore, both materials show better miscibility for smaller silica particle sizes. For lower mass ratios, the PSD resembled the PSD of pure carbon black after the spray flame with one mode (Figure 5b). Figure 6b shows the dependency of the first mode of particle size on the mass ratio of the silica particles. Due to the formed hetero-contact, by sintering silica particles onto the carbon black aggregate structure, an increase in the mode hydrodynamic aggregate diameter is expected, as the sedimentation speed is influenced by the aggregate density, which may be increased by the attached silica ($\rho_{\mathrm{CB}}=1.86\;\frac{g}{{\mathrm{cm}}^{3}}$ [42]; $\rho_{\mathrm{SiO}2}=2.1\;\frac{g}{{\mathrm{cm}}^{3}}$ depending on porosity [43]). Therefore, the hydrodynamic aggregate size is a measure for the effectiveness of the incorporation of silica into existing CB structures. For all particle sizes, the largest mode value is observed for an equal mass ratio of silica to carbon black, showing the optimal incorporation of the silica in the carbon black structure. For the hetero-aggregates produced with a silica particle size of 60 nm, the least pronounced change in the hydrodynamic aggregate size is observed with a spread between 218 ± 19 nm and 255 ± 8 nm. Furthermore, in Figure 6c, a TEM image of a hetero-aggregate produced with 60 nm silica particles is depicted. Therein, a carbon black aggregate exhibits only one visible silica hetero-contact despite a mass ratio of silica to carbon black of 3, supporting the drawn hypothesis of increasing the segregation of the two materials for larger particle sizes of silica. Furthermore, the change in the hydrodynamic aggregate size for varying mass ratios of silica was the most pronounced for hetero-aggregates produced with 10 nm silica particles. Here, the maximum measured hydrodynamic aggregate size was 357 ± 10 nm. Figure 6a underlines the influence of silica particle size on the size of the hydrodynamic aggregate. In the EDXS scan, a homogenous distribution of silica on the carbon black structure is apparent, meaning that both materials were successfully hetero-aggregated. The decreasing mode aggregate diameter for mass ratios larger than 1 independent of silica particle size indicates segregation due to the homo-aggregation of silica. However, in all cases, the mode of the hetero-aggregates was larger than for the pure carbon black in the spray flame. This means in all cases, some degree of hetero-aggregation was achieved. 

In conclusion, the data retrieved via both ADC and SAXS suggest that optimal hetero-aggregation is achieved for mass ratios of silica to carbon black equal to or below 1 and particle sizes of silica in the size range of 10 nm. This assumption is supported by the USAXS data, which reveal a fractal dimension of mass of 2.2, which is equal to the aggregates of pure carbon black (Figure 3a). The increased hydrodynamic aggregate size of 258 nm in comparison to pure carbon black after the spray flame with 201 nm shows successful hetero-aggregation. For higher mass ratios, increased segregation due to the homo-aggregation of silica is observed. 

As an extra finding, a layer of carbon was formed on the surface of the silica particles in the spray flame, as revealed in Figure 7. Although this was not observed for all surveyed aggregates, the carbon layer is expected to increase the overall conductivity of the aggregate and is therefore desirable. 

## 4. Discussion

Carbon black is a widely used additive for to improve conductivity in batteries. Therein, binders like PVDF ensure the binding of the active material as well as the stabilization of the electric pathways of carbon black. However, the environmental hazards of PVDF motivate the search for alternatives. A novel hetero-aggregation process is presented with the goal of improving the inherent mechanical strength and dispersibility of carbon black by incorporating colloidal silica into the carbon black structure via sintering in a spray flame. This work focuses on a comprehensive description of hetero-aggregate properties via multi-scale characterization through the use of SAXS, HAADF-STEM with EDXS and ADC. Therein, the particle size and mass ratio of the silica particles were identified as main factors in successful hetero-aggregation, which is achieved when a sinter contact is formed without significant changes in primary particle properties. In the evaluated TEM images, the majority of silica particles preserved their spherical shape and original diameter while forming aggregates connected by sinter bridges. Therefore, it is suggested that the surface of the silica is partially melted upon particle–particle contact which, in turn, leads to plastic deformation, forming a contact area. The negligible changes in the primary particle properties can be attributed to the short flame residence times of ≈0.7 ms [44]. However, previous researchers suggested the existence of a toroidal vortex within the spray flame of the SpraySyn burner [45]. Recirculation within this vortex may explain the infrequent observed coalescence of silica particles. The coalescence of silica particles has been reported for temperatures ranging from 1300 to 1700 K [46,47], which are similar to the gas phase temperature of the SpraySyn burner (around 1500 K in the spray flame [34]) used in this study. The rate of coalescence is directly influenced by the particle size, leading to increased coalescence rates for smaller particles [48]. However, the observed segregation of silica and carbon black for 60 nm silica particles resulted in an increased number of silica homo-contacts which, in turn, facilitated the creation of homo-aggregates. A similar trend was observed for mass ratios of silica to carbon black above 1. The increased concentration led to an increased particle collision frequency which, in turn, increased the coalescence rate [49] and the formation of homo-aggregates. The results of the different measurement systems are consistent with each other. In the USAXS data, the formation of homo-aggregates is identified by an increased exponent in the power law fit, leading to values of up to 2.8 in comparison to the original carbon black used (1.8–2.2). Additionally, in the ADC measurements, the homo-aggregates appear as a second mode in the micron size range. 

In conclusion, larger silica particles may deflect upon contact with the carbon black aggregates due to their higher mass, whereas smaller silica particles tend to adhere and sinter to the carbon black. Therefore, the results suggest that the optimal experimental conditions for the hetero-aggregation of carbon black and silica are mass ratios equal to or below one and low silica primary particle sizes (10 nm). Smaller silica particle sizes are only achievable via a modification of the Stoeber method, e.g., through the addition of Triton X-100, which would lead to impurities in the spray flame [50]. TEM images of the hetero-aggregates produced under these experimental conditions show a homogeneous distribution of the silica on the carbon black structure. Furthermore, a fractal dimension of mass close to the original value for carbon black (2.2) and an increased hydrodynamic aggregate diameter of 258 nm were measured, which further supports the assumption of high-quality hetero-aggregation. Additionally, individual silica particles coated with carbon layers were identified. They are expected to exhibit advantageous properties for Li-ion battery applications [12,51] and should be investigated in further studies. In summary, the obtained results emphasize the general applicability of spray flames for the production of novel hetero-aggregate materials. Future studies will focus on the determination of functional properties like conductivity, dispersibility and the application of these hetero-aggregates in batteries.

## Figures and Tables

**Figure 1 nanomaterials-13-01893-f001:**
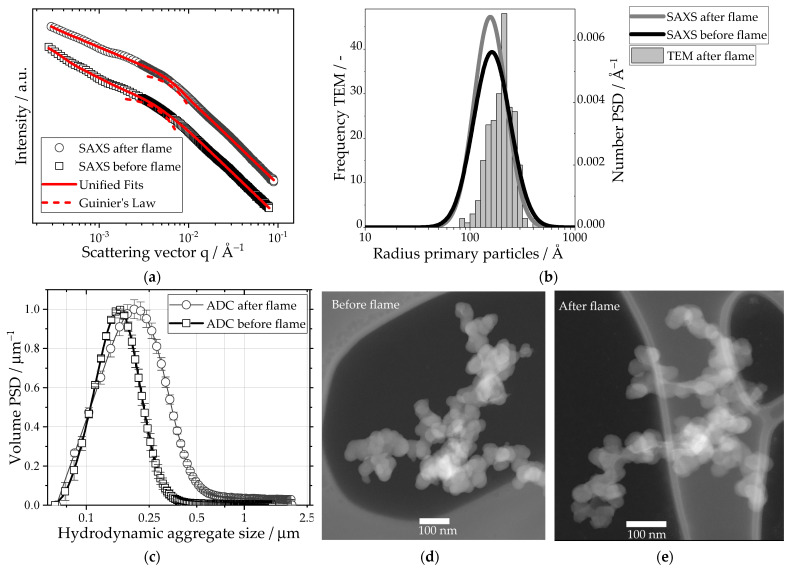
(**a**) SAXS data of carbon black before and after the spray flame. The original scattering data has been fitted with a unified fit. The local Guinier fit of the primary particles is highlighted with a dashed line. (**b**) Size distributions calculated from the fit parameters of the unified fit. Additional TEM data analysis of 224 counted primary particles of the carbon black after the spray flame yields a comparative size distribution. (**c**) Normalized volume PSDs of the carbon black aggregates before and after the spray flame, obtained via ACD, yield a hydrodynamic diameter. (**d**) HAADF-STEM image of the carbon black before the spray flame and (**e**) after the spray flame.

**Figure 2 nanomaterials-13-01893-f002:**
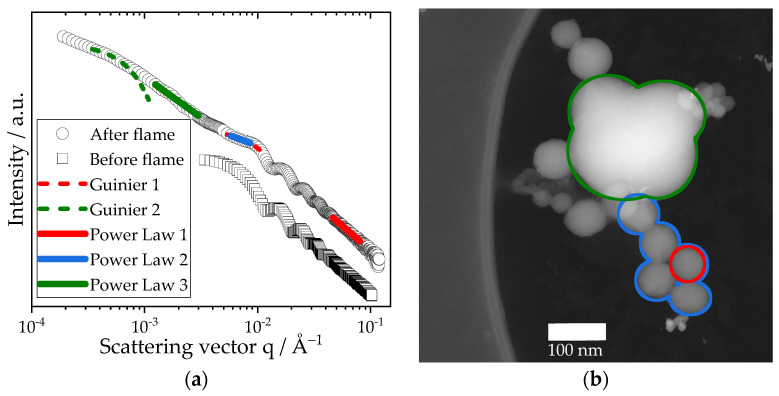
(**a**) SAXS data of the investigated colloidal silica particles before and after the spray flame. The scattering curve of the particles was fitted with local power law and Guinier fits. The colours of the local fits correspond to the colours of the structures formed, which are marked in the HAADF-STEM image of the sample after the spray flame in (**b**). The sample before the flame was measured as a dilute suspension.

**Figure 3 nanomaterials-13-01893-f003:**
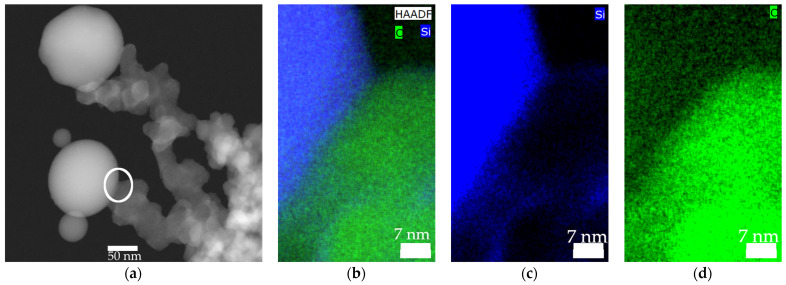
(**a**) HAADF-STEM image of a hetero-aggregate with visible hetero-contact (white circle) (**b**–**d**) Closeup HAADF-STEM image with EDXS of the hetero-contact, with carbon (symbol C) coloured in green and silicon (symbol Si) coloured in blue. Experimental conditions were a mass ratio of silica to carbon black of 3 and original colloidal silica primary particle size of 30 nm.

**Figure 4 nanomaterials-13-01893-f004:**
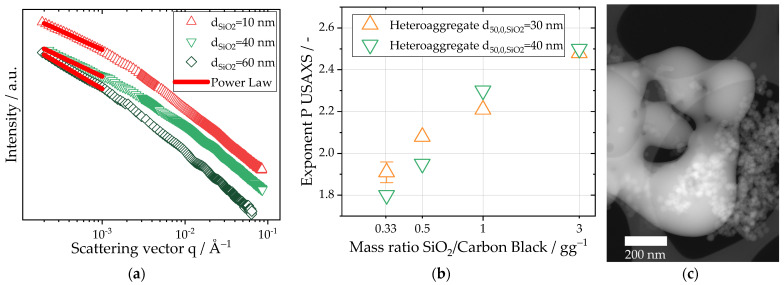
(**a**) SAXS data of hetero-aggregates produced at a mass ratio of silica to carbon black of 1 with varying sizes of the colloidal silica particles ranging from 10 nm to 60 nm. The evaluated power law fit in the USAXS area is highlighted in red. (**b**) Derived exponents of the local power law fit in the USAXS area for hetero-aggregates with different mass ratios of silica to carbon black ranging from 0.3 to 3. Two different primary particle size ranges of the colloidal silica were investigated: 30 nm and 40 nm. In order to evaluate the experimental spread, three individual experiments were evaluated for a mass ratio of 0.33. (**c**) Observed homo-aggregation and coalescence of silica particles for a mass ratio of SiO_2_ to carbon black of 3 and 30 nm silica particles in HAADF-STEM images.

**Figure 5 nanomaterials-13-01893-f005:**
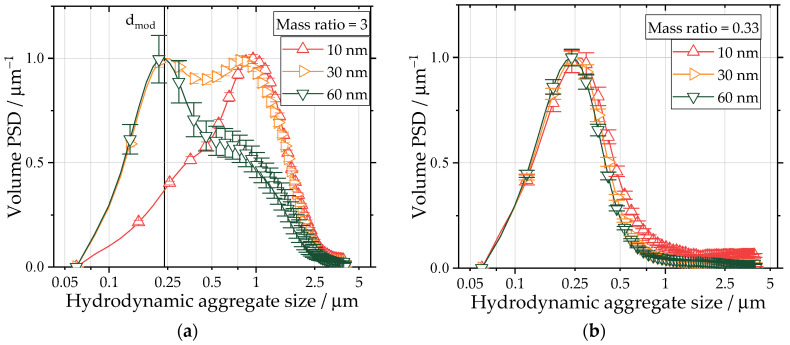
(**a**,**b**) Normalized volume PSDs of hetero-aggregates, as measured via ADC. For better visual clarity, only every fifth data point is shown. The investigated hetero-aggregates were produced with mass ratios of silica to carbon black of 3 (**a**) and 0.33 (**b**) for 10 nm, 30 nm and 60 nm colloidal silica particles, respectively.

**Figure 6 nanomaterials-13-01893-f006:**
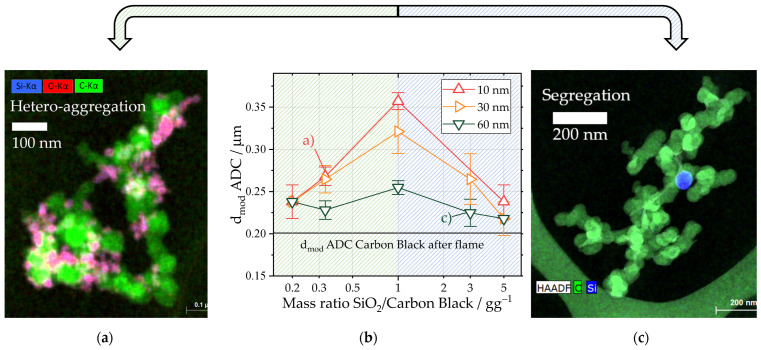
(**a**,**c**) HAADF-STEM images with EDXS of a hetero-aggregate produced with (**a**) 10 nm silica particles at a mass ratio of 0.33 and (**c**) 60 nm silica particles at a mass ratio of 3. Carbon is coloured in green (symbol C), silicon in blue (symbol Si) and oxygen in red (symbol O). Therefore, the silica (SiO_2_) particles in (**a**) show a pinkish colour. (**b**) First mode (highlighted in Figure 5a with a straight line) of the PSD of hetero-aggregates for mass ratios of silica to carbon black ranging from 0.2 to 5 for three silica particle sizes. The data points each represent two experiments with three respective ADC measurements. The highlighted data points (**a**,**c**) correspond to the TEM images depicted in (**a**,**c**).

**Figure 7 nanomaterials-13-01893-f007:**
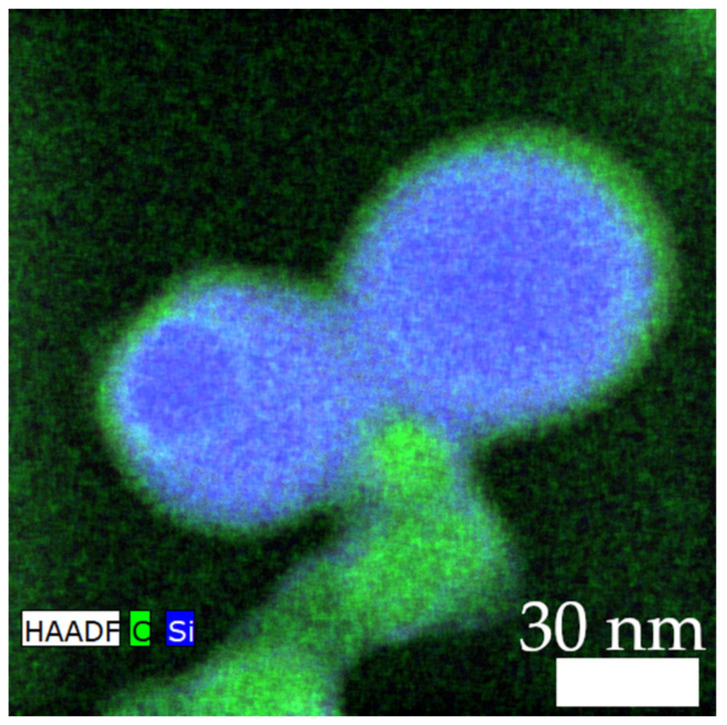
Carbon layer on the surface of the silica particles (diameter of silica = 60 nm; mass ratio = 3). Carbon is coloured in green (symbol C) and silicon in blue (symbol Si).

**Table 1 nanomaterials-13-01893-t001:** Overview of the colloidal silica particles produced and their mean diameters.

Name	$d_{50,0}\;\mathrm{in}\;\mathrm{nm}$
60 nm	63.6 ± 14.9
40 nm	39.2 ± 11.9
30 nm	32.0 ± 8.1
10 nm	11.6 ± 3.4

## Data Availability

Data will be made available upon request.

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
