# Peer review of "Spray Flame Synthesis and Multiscale Characterization of Carbon Black–Silica Hetero-Aggregates"

_nanomaterials, 2023, doi:10.3390/nano13121893_

Round 1

Reviewer 1 Report

In this study, the authors presented a method for structuring carbon black aggregates by adding colloidal silica in a spray flame 10 with the goal of opening up more choices for polymeric binders. The manuscript was well written in general. However, before publication, following points should be considered:

1.      An illustration of the sample preparation or at least several pictures are highly suggested, otherwise it is hard to get what have you done at the very first glance.

2.      I suggest all the (a)(b)(c) should be presented on the figure, rather not listed below the figures.

3.      Line 379. Since the conclusion indicates the optimal diameter of the silica particle should be 10 nm, then you have to add several sentences to explain the design of the experiment. Because you have listed the minimal diameter of silica of 10 nm. Then it is suspicious that 5 nm might achieve a better result.

4.      The difference between Figure -d and -e should be highlighted. I don’t see any valuable information from the current both figures. Plus, you did not even mention the figure 1e in the context at all.

Reviewer 2 Report

The authors used spray flame method synthesis carbon-silica hetero-aggregates. It is a scalable method and may have potential applications. However, the following questions should be well addressed before further consideration.

1. The multiscale characterization focus on the particle size and aggregate state of carbon-silica. What kind of structure can help improve the performance of lithium-ion batteries? Relevant references must be cited.

2. Line 237, the phrases “Please note that” should be deleted.

3. The word “publication” in line 9 and 346 is not proper.

4. Line 344, the sentence “PVDF improve the dispersibility of carbon black and ensure electric pathways” is confusing. How an insulated binder improves the dispersibility and the conductivity of the active materials.

5. What is the meaning of the mechanical strength and dispersibility of carbon black with the colloidal silica? Please provide quantified proof that this structure can replace the PVDF binder in a lithium-ion battery.

Minor editing of English language required
